# Direct Current Electrical Stimulation Shifts THP-1-Derived Macrophage Polarization towards Pro-Regenerative M2 Phenotype

**DOI:** 10.3390/ijms25137272

**Published:** 2024-07-02

**Authors:** Santiago Bianconi, Liudmila Leppik, Elsie Oppermann, Ingo Marzi, Dirk Henrich

**Affiliations:** 1Department of Trauma Surgery and Orthopedics, University Hospital, Goethe University Frankfurt, 60590 Frankfurt am Main, Germany; leppik@med.uni-frankfurt.de (L.L.); marzi@trauma.uni-frankfurt.de (I.M.); d.henrich@trauma.uni-frankfurt.de (D.H.); 2Department of General, Visceral, Transplant and Thoracic Surgery, University Hospital, Goethe University Frankfurt, 60590 Frankfurt am Main, Germany; oppermann@em.uni-frankfurt.de

**Keywords:** direct current electrical stimulation, macrophage polarization, THP-1-derived macrophages, tissue regeneration

## Abstract

A macrophage shift from the M1 to the M2 phenotype is relevant for promoting tissue repair and regeneration. In a previous in vivo study, we found that direct current (DC) electrical stimulation (EStim) increased the proportion of M2 macrophages in healing tissues and directed the balance of the injury response away from healing/scarring towards regeneration. These observations led us to hypothesize that DC EStim regulates macrophage polarization towards an M2 phenotype. THP-1-derived M0, M1 (IFN-γ and LPS), and M2 (IL-4 and IL-13) macrophages were exposed (or not: control group) to 100 mV/mm of DC EStim, 1 h/day for three days. Macrophage polarization was assessed through gene and surface marker expressions and cytokine secretion profiles. Following DC EStim treatment, M0 cells exhibited an upregulation of M2 marker genes IL10, CD163, and PPARG. In M1 cells, DC EStim upregulated the gene expressions of M2 markers IL10, TGM2, and CD206 and downregulated M1 marker gene CD86. EStim treatment also reduced the surface expression of CD86 and secretion of pro-inflammatory cytokines IL-1β and IL-6. Our results suggest that DC EStim differentially exerts pro-M2 effects depending on the macrophage phenotype: it upregulates typical M2 genes in M0 and M1 cells while inhibiting M1 marker CD86 at the nuclear and protein levels and the secretion of pro-inflammatory interleukins in M1 cells. Conversely, M2 cells appear to be less responsive to the EStim treatment employed in this study.

## 1. Introduction

Macrophages are heterogeneous and versatile cells that are part of the innate immune system. They are widely distributed across all body tissues, playing a multitude of functions depending on the tissue niche in which they reside [1,2]. Based on environmental cues, macrophages undergo significant phenotypic and functional changes that can be broadly categorized into M1 (pro-inflammatory) and M2 (anti-inflammatory) states [3,4]. This polarization state is not fixed, as macrophages possess sufficient plasticity to integrate multiple signals from sources such as microbes, damaged tissues, and the normal tissue environment [5].

Macrophages play a pivotal role throughout the initiation, maintenance, and resolution phases of tissue repair. They orchestrate the activities of several players like fibroblasts, epithelial cells, and stem cells while regulating the body’s immune response (reviewed in [6]). Experimental studies have shown that removing macrophages impairs both healing and regeneration [7,8,9,10]. Following injury, macrophages initiate inflammation and facilitate the clearance of cellular debris, pathogens, and damaged tissues. During the healing process, they actively contribute to histolysis, reepithelialization, revascularization, extracellular matrix (ECM) synthesis, and cell proliferation. In the later stages of tissue repair, macrophages help resolve inflammation and contribute to tissue remodeling (reviewed in [6]).

The balance between M1 and M2 macrophages is crucial for proper tissue healing. During the initial stages of the healing process, M1 macrophages dominate the scene, while the proportion of M2 macrophages increases over time. This shift towards the M2 phenotype is important for promoting tissue repair and regeneration, as M2 macrophages dampen the initial inflammation and aid in tissue remodeling, angiogenesis, and the formation of new ECM components (reviewed in [11,12]). Therefore, increasing the ratio of M2 to M1 macrophages has been proposed as a potential strategy to improve tissue remodeling outcomes [13]. For instance, in the context of bone healing, promoting M2 macrophage polarization has been observed to significantly enhance fracture repair [14,15,16].

Bioelectric signaling has been shown to play a significant role in tissue regeneration (reviewed in [17,18,19]). Changes in the cell membrane potential can significantly modulate cell functions such as the differentiation and transdifferentiation capacities of mesenchymal stem cells (MSCs) [20,21]. Previous studies conducted in our laboratory, for example, demonstrated that external direct current (DC) electrical stimulation (EStim) can enhance the osteogenic differentiation potential of MSCs in vitro [22,23]. Furthermore, the tissue inflammatory response, which can extensively modify the healing outcome, can also be regulated using external EStim (reviewed in [24]), as demonstrated in the context of bone injury [25].

Macrophage activity can be modulated not only by chemical factors but also by physical stimuli, such as electricity, within a complex microenvironment. Cell migration, phagocytic uptake, and cytokine production have been reported to be modified using EStim in in vitro experiments [2]. Animal studies have demonstrated that the application of external EStim upregulates M1 macrophages in the inflammatory phase of wound healing and also positively affects the proliferative and remodeling phases thereafter [26]. In a previous study performed in our laboratory, DC EStim treatment during tissue healing in a rat limb amputation model led to a sustained increase in the pro-regenerative M2 cell count. This was accompanied by modifications in the extracellular matrix structure and gene expression, suggesting that the EStim treatment shifted the injury response balance from healing/scarring towards regeneration [27]. Based on these observations, we hypothesized that DC EStim can induce either the M2 polarization of unpolarized cells and/or transdifferentiation of M1 cells into the M2 phenotype.

Although macrophages are known to play a crucial role in tissue regeneration, and their responsiveness to EStim has been demonstrated, the potential effects of DC EStim on macrophage polarization and, consequently, on tissue regeneration, remain largely unexplored. In this study, we aimed to assess the in vitro capacity of DC EStim to regulate macrophage polarization. To achieve this goal, we examined the combined effects of DC EStim and cytokine exposures on THP-1-derived macrophages during M1 (IFN-γ and LPS) and M2 (IL-4 and IL-13) polarization, as well as the effects of DC EStim on non-polarized macrophages (M0).

## 2. Results

### 2.1. Induction of THP-1 Differentiation and Polarization

After the priming of THP-1 cells with 5 ng/mL phorbol 12-myristate 13-acetate (PMA) for 24 h, the cells became adherent. After exposure to cytokines, the M1 and M2 cells developed irregular shape, with multiple extensions (pseudopodia) protruding from the cell body (Figure 1).

The macrophage response to M1- and M2-polarizing treatments was confirmed through the assessment of typical M1 and M2 surface markers, gene expressions, and cytokine secretion profiles (data corresponding to the control groups are depicted in the figures below). In the three phenotypes studied here, the LPS receptor CD14 and the integrin CD11b were strongly expressed, indicating successful differentiation in response to PMA stimulation and the effect of a resting time. At the nuclear level, the treatment with IFN-γ + LPS induced the gene expressions of M1 markers CD86, TNF, IL1B, IL6, IL8, NFKB1 (nuclear factor kappa B subunit 1), STAT1 (signal transducer and activator of transcription 1), and IRF5 (interferon regulatory factor 5) alongside a significant downregulation of M2 marker CD206 in M1 compared to the M0 control (*p* < 0.05). Conversely, exposure to IL-4 + IL-13 upregulated the gene expressions of M2 markers PPARG (peroxisome proliferator-activated receptor gamma), TGM2 (transglutaminase 2), and SOCS1 (suppressor of cytokine signaling 1) in M2 compared to the M0 control (*p* < 0.05) but did not affect other M2 markers.

As expected, the activation with IFN-γ +LPS induced a significant increase in the secretion of the pro-inflammatory cytokines IL-1β, IL-6, IL-8, and TNF-α in the M1 phenotype. In contrast, in the M0 and M2 phenotypes, only IL-8 secretion was increased after the exposure to priming or priming and polarizing stimuli, respectively, albeit in a very modest manner compared to the robust M1 response, while the levels of the other cytokines remained undetectable. Interestingly, we did not detect any increase in the secretion of the typical M2 cytokine IL-10 after three days of exposure to IL-4 and IL-13.

### 2.2. DC EStim Does Not Affect Cell Count and Viability of M0, M1, and M2 Macrophages

To analyze the impact of EStim on cell health, we first assessed whether DC EStim had an effect on the cell number (Figure 2A) and viability (Figure 2B) of differentiated macrophages at day 3 of polarization. We observed that the EStim treatment did not produce any change in the cell counts in M0, M1, and M2 phenotypes compared to the respective control cells (*p* > 0.05). From the comparison between M0, M1, and M2 control groups, a significantly higher count of M1 than M2 cells was detected (*p* < 0.05), while no differences were observed between these groups and the M0 one (*p* > 0.05). Similarly, no effects of EStim were observed on the viabilities of M0 and M1 cells. Conversely, an increase in the reduction of alamarBlue was detected in the M2 EStim vs. M2 control group (*p* < 0.05). Additionally, the M1 control showed a higher viability than M0 and M2 controls (*p* < 0.05).

### 2.3. DC EStim Reduces Expressions of Cell Surface Markers CD14 in M1 and CD86 in M1 and M2 Macrophages

The expression of cell surface markers in THP-1-derived macrophages was analyzed using flow cytometry (Figure 3 and Appendix A). The percentage of CD14+ and CD86+ cells was significantly lower in the M1 group exposed to EStim during polarization (*p* < 0.05) compared to its control, whereas no differences in CD11b+ and CD80+ cells were detected. In M0 cells, EStim did not produce any changes in the percentage of CD14+ and CD11b+ cells. The expressions of CD14, CD11b, and CD206 were not affected by the EStim treatment in M2 cells.

### 2.4. DC EStim Downregulates CD86 and IL1B Gene Expressions in M1 Cells While Upregulating Expressions of M2 Phenotype Markers in M0 and M1 Macrophages

Gene expression analysis of M1 and M2 phenotype markers was performed by means of RT-qPCR. EStim treatment downregulated the CD86 and IL1B gene expressions in M1 macrophages (*p* < 0.05) (Figure 4). Furthermore, EStim upregulated IL1B, IL8, IL10, CD163, and PPARG in M0 cells and IL8, IL10, TGM2, and CD206 in M1 cells (*p* < 0.05) (Figure 4 and Figure 5). The rest of the M1- and M2-related genes did not significantly differ when comparing the control and experimental groups.

### 2.5. DC EStim Reduces Secretion of IL-1β and IL-6 in M1 Cells While Increasing Secretion of IL-8 in M0, M1, and M2 Cells

To assess the potential influence of DC EStim on the macrophage secretome in vitro, we measured cytokine concentrations in the cell supernatant. The IL-1β and IL-6 levels were reduced in the M1 EStim group, and the IL-8 concentration showed higher values in all the groups exposed to EStim (*p* < 0.05) compared to their respective controls (Figure 6). TNF secretion increased with LPS/IFN-γ treatment, but there were no significant differences in this parameter between M1 control (169.11 ± 8.18 pg/mL) and M1 EStim (174.16 ± 6.06 pg/mL) cells.

## 3. Discussion

The present study aimed to verify whether DC EStim is able to support macrophage polarization towards the anti-inflammatory and pro-healing M2 phenotype. Our results show that DC EStim, when applied to M0 cells, upregulates the typical marker genes of M2 polarization such as IL10, CD163, and PPARG. Furthermore, a pro-M2 effect was also observed when DC EStim was applied to M1-polarizing cells, characterized by an upregulation of M2 marker genes IL10, TGM2, and CD206, downregulation of CD86 and IL1B gene expressions, lower expression of surface protein CD86, and reduced secretion of pro-inflammatory cytokines IL-1β and IL-6. To the best of our knowledge, there have been no previous reports of a DC EStim M2-polarizing effect in M0 cells in the absence of polarizing cytokines.

In a previous study performed in a rat limb amputation model, we found that DC EStim treatment was able to shift the injury response from healing/scarring towards regeneration. This was demonstrated by significant changes in the ECM deposition, which included less condensed collagen fibrils and the activation of genes related to morphogenesis and development. Additionally, the electrically stimulated stump tissues showed a significant increase in the M2/M1 macrophage ratio [27]. However, based on these results, it was not possible to elucidate whether EStim directly promotes M2 polarization or if its effects were indirect, such as reducing the M1 population through cytotoxicity.

The phenotypic plasticity of macrophages allows them to respond dynamically to changing conditions [28]. Furthermore, once polarized, macrophages can undergo a reprogramming process of transdifferentiation from M1 to M2 or vice versa [29]. To better understand the macrophage-polarizing response to DC EStim, we employed the human monocytic cell line THP-1, recognized as a suitable model for the investigation of relatively straightforward biological processes such as polarization and its functional implications [28]. This cell line responds to activating stimuli similarly to human monocyte-derived macrophages (MDMs), particularly in the expressions of typical nuclear markers and the biological activities of their conditioned media on other cell lines. However, differences between THP-1-derived macrophages and MDMs have been reported in terms of the intensity of gene expression (more pronounced in MDMs) and protein secretion (including cytokines, chemokines, and growth factors) in response to polarizing stimuli [28,30]. For instance, the expression of the M2 protein marker CD163 was undetectable in THP-1-derived macrophages [28], as confirmed in our preliminary experiments. Moreover, MDM culture conditions and their responses to stimuli better emulate the in vivo macrophage conditions. Therefore, future studies including MDMs should be conducted to explore the effects of DC EStim within a more complex and realistic range of macrophage responses.

Like other cells in the body, immune cells are influenced by the application of EStim, as demonstrated by numerous in vitro and in vivo studies. EStim has been shown to modulate the activation and proliferation of T and B cells [31,32], as well as the migration of neutrophils [33], lymphocytes [34,35], and macrophages [2,36]. Additionally, EStim affects macrophage phagocytic uptake and cytokine secretion [2] and systemic cytokine release [37]. These findings underscore EStim’s potential as a noninvasive method to modulate immune cell phenotypes and activities, which could be beneficial for treating chronic wounds, autoimmune diseases, and various forms of cancer [38].

The effect of EStim on macrophage polarization has been extensively explored, demonstrating its ability to support either M1 or M2 polarization depending on factors such as the type and regimen of EStim, the presence and doses of polarizing agents, the polarization stage at the time of EStim treatment, the cell source, and the methods used to assess polarization [38]. For instance, Gu et al. (2022) treated mouse bone marrow-derived macrophages with alternating current pulsed stimulation for 6 h daily, varying the voltages, frequencies, and waveforms alongside polarizing cytokines. They found that EStim with a square waveform promotes M1 polarization, while EStim with a sinusoidal waveform enhances cytokine-induced macrophage polarization into both M1 and M2 phenotypes [39]. This variability underscores the need for further research to elucidate the underlying mechanisms and optimize stimulation protocols for desired therapeutic outcomes.

Among others, EStim can regulate a cell’s metabolic activity [23,40] and adherence capacity [41,42], both of which are functions known to be relevant for tissue repair. The results of our study show that DC EStim treatment applied during macrophage polarization does not reduce the cell number and viability and, conversely, in the M2 group, increased the percentage of reduction of alamarBlue, which could be attributed to a higher metabolic activity. In general, EStim can make cells more active by increasing their metabolism [43]. Furthermore, macrophages can undergo metabolic reprogramming according to their polarization state. Typically, M1 macrophages exhibit enhanced glycolysis with reduced oxygen consumption, whereas M2 cells show increased fatty acid oxidation and oxidative phosphorylation, maintaining an intact Krebs cycle (reviewed in [44]). The alamarBlue assay evaluates the overall redox state of the cells, which is influenced by various metabolic pathways including glycolysis, the citric acid cycle, and other cellular processes [45]. Hence, it was not possible to discriminate whether the changes in the metabolic activity were caused by the activation of oxidative or non-oxidative pathways and, consequently, to determine if these changes were associated with the M2 or M1 metabolic profile.

Another in vitro study employing a similar DC EStim device and regimen but murine macrophage cell line (J774A.1; 91.4% M1 phenotype) reported a decreased culture capacity to reduce resazurin after cell treatment with EStim [40]. The authors also assessed the influence of faradaic products by exposing the cells to electrically stimulated acellular media, and they reported the same effect. In our experimental setting, where the medium was not replaced during the 3-day EStim period, neither EStim nor faradaic products produced this negative effect on cell viability. Furthermore, unlike the aforementioned study, we did not observe a lower cell density in the vicinity of the electrodes in the cell culture. This could be attributed to a higher resistance of THP-1 cells to the current density around the electrodes compared to J774A.1 macrophages.

THP-1-derived macrophages express typical surface markers after priming and polarization into specific phenotypes [46]. In the M1 group, DC EStim treatment reduced the percentage of CD14+ cells. This finding may suggest that EStim reverts the PMA differentiating effect on M1 cells, making them less responsive to LPS, which was one of the polarizing stimuli used in the present study. Whether this effect is definitive or reversible should be elucidated in future studies, for example, by exposing cells again to PMA.

The markers CD80 and CD86 (B7-1 and B7-2, respectively) are suitable to identify the M1 subpopulation of macrophages (reviewed in [47,48,49]). DC EStim exposure during polarization drastically reduced the expression of the CD86 protein in M1 cells, while the expression of the CD80 protein remained intact. At the gene expression level, a significant downregulation of CD86 in the M1 EStim group was also observed. The lower percentage of cells expressing the surface LPS receptor CD14 in the M1 EStim group could be one of the reasons behind this change. Nevertheless, the current study does not allow us to determine the order of the events, and further studies should be conducted to test this hypothesis.

The members of the M2 transcriptome assessed in the present study are sensors or effectors involved in M2 immunomodulatory properties (reviewed in [3]). Exposure to DC EStim upregulated the gene expressions of M2 markers IL10, CD163, and PPARG in M0 cells and IL10, TGM2, and CD206 in M1 macrophages. This suggests that at the transcriptional level, DC EStim induces the polarization of M0 cells and transdifferentiation of M1 cells towards the M2 phenotype. Furthermore, signaling mechanisms involving PPARG can be postulated to play a role in the M2 polarizing effect induced by EStim on M0 cells [50]. However, the participation of M2 mediators and effectors, such as STAT6 and SOCS1, could not be confirmed in the present study at the RNA level. Therefore, the pathways responsible for the transdifferentiating effect on M1 cells should be further explored.

Macrophages activated by chemical mediators, such as IFN-γ, LPS, or IL-4, adopt strong functional phenotypes that produce different cytokine-release patterns [2]. In the present study, DC EStim selectively modulated cytokine production in a phenotype-dependent way, reducing M1 transcription and the secretion of IL-1β and secretion of IL-6 while increasing IL-8 secretion in M0, M1, and M2 cells. A differential phenotypic response to EStim was also observed in the study by Hoare et al. (2016), where the treatment with EStim increased TNF-α and neurotrophin-3 secretion in M0 cells and IL-1β and IL-23 in M1 cells, with no influence in cytokine secretion in M2 cells [2].

The inflammatory response induced by mediators like TNF-α, LPS, and IL-1β can trigger or exacerbate tissue damage and delay the healing process [51]. Specifically, the sustained production of IL-1β by macrophages has been shown to be a major driver of persistent inflammation and fibrosis [6]. Therefore, the observed changes induced by DC EStim in the M1 secretion of IL-1β and IL-6 could positively influence not only the macrophage functions but also the overall healing process. For example, the inhibition of IL-1β has been reported to result in improved wound healing via the induction of a reparative macrophage phenotype [52]. Furthermore, macrophage IL-6 production can be boosted by TNF-α and IL-1β [53]. This interrelationship could explain the simultaneous reduction of both IL-1β and IL-6 secretion with the EStim treatment. In line with these findings, a previous in vivo study performed in rats also showed that EStim can reduce the production of pro-inflammatory cytokines during bone regeneration and also improve angiogenesis and osteogenesis [25].

Regarding the effect of EStim on IL-8, we observed an upregulation of IL8 (CXCL8) gene expression in M0 and M1 phenotypes and higher secretion of IL-8 in all three phenotypes after treatment with DC EStim. IL-8 is a chemotactic factor that plays relevant roles during the inflammatory response [54,55]. Furthermore, it is also known to be a potent promoter of angiogenesis [56], which could explain, for example, the increase in vascularization after exposure to EStim in the context of stump tissue healing [57] or bone healing [25]. The tissue regeneration process is characterized by an initial increase in the local stem cell populations and ECM remodeling, followed by new vessel growth [58]. This proangiogenic effect of EStim could be relevant in the field of tissue repair, especially in bone tissue engineering, which faces challenges such as a lack of sufficient vascularization at the defect site [59].

M1 cytokine production is primarily regulated by the activation and nuclear translocation of the transcription factor NF-κB (nuclear factor kappa-light-chain enhancer of B cells), together with STAT1, STAT3, IRF5, HIF1α (hypoxia-induced factor 1 alpha), and AP1 (activator protein 1) (reviewed in [44]). In the present study, NF-κB, STAT1, and IRF5 were found to be upregulated after exposure to M1-polarizing stimuli. However, no significant changes were detected in the treated groups compared to their respective control. This suggests that DC EStim might influence IL-1β, IL-6, and IL-8 secretion through alternative signaling mechanisms, highlighting the complexity of cytokine regulation in immune responses.

## 4. Materials and Methods

### 4.1. Cell Culture

THP-1 cells were purchased from the European Collection of Authenticated Cell Cultures (TIB-202, ECACC) and used as received. The cells were cultured in growth medium (GM, 10% fetal bovine serum (Gibco, Paisley, UK), 100 U/mL penicillin, and 100 µg/mL streptomycin (Sigma-Aldrich, St. Louis, MO, USA) in RPMI 1640 medium with L-glutamine (Gibco, Paisley, UK) at 37 °C in a humidified 5% CO_2_ atmosphere. The cells were allocated into 6 groups (Table 1) and treated for 5 days according to the experimental protocol outlined in Figure 7 and described subsequently. On experimental day 5, cell supernatant was collected, cleared using centrifugation (300× *g*, 5 min), and stored at −80 °C for the subsequent assessment of cytokine secretion profiles. The cells were washed twice with sterile PBS without Ca^2+^Mg^2+^ (Gibco, Paisley, UK) to remove non-adherent cells. The cell count, viability, expressions of cell surface markers, and gene expression were assessed in adherent cells.

### 4.2. Cell Priming and Polarization

To prime THP-1 cells into macrophage-like cells, 7 × 10^5^ THP-1 cells in 2 mL of GM were seeded into a well of a 6-well plate (TPP, Trasadingen, Switzerland) and cultured for 24 h in the presence of 5 ng/mL phorbol 12-myristate 13-acetate (PMA, Sigma-Aldrich, Taufkirchen, Germany) (Figure 7). This concentration was found to be sufficient to induce stable differentiation without undesirable gene upregulation [60]. Following incubation, the medium and non-adherent cells were completely removed, and adherent cells were washed twice with sterile PBS without Ca^2+^Mg^2+^. A total of 2 mL of fresh GM were added to each well, and after a resting period of 24 h, the cells in the M1 and M2 groups (control and experimental groups) were polarized by incubation for 72 h in GM supplemented with 20 ng/mL IFN-γ (R&D Systems, Minneapolis, MN, USA) and 100 ng/mL LPS (Cat Num L2630, Sigma-Aldrich, Taufkirchen, Germany) or 20 ng/mL IL-4 and 20 ng/mL IL-13 (both from R&D Systems, Minneapolis, MN, USA), respectively. Cells in the M0 control and experimental groups were cultured in GM for 72 h without any polarizing agent [28,29,61]. Polarized cells were observed, and phase contrast images were captured using an Axioobserver Z1 microscope (Zeiss, Gottingen, Germany). The timing of the priming and resting periods prior to cytokine exposure was optimized by testing the combinations of 24−48, 48−24, and 24−24 h, respectively. The latter showed higher percentages of CD14+ (97%) and CD11b+ (85%) cells.

### 4.3. Direct Current Electrical Stimulation

Cells from the experimental groups M0 EStim, M1 EStim, and M2 EStim were exposed to 100 mV/mm of DC EStim for 1 h per day for three days, starting from experimental day 2 using a custom-made DC EStim device consisting of a 6-well cell culture plate lid provided with 1 pair of platinum electrodes (35 mm long, 1 mm diameter) per well, bent into an L-shape and positioned 25 mm apart from each other. During the EStim treatment, the ends of the electrodes were completely immersed in the culture medium in a horizontal plane 4 mm above the cells (Appendix A) [62,63,64]. For sterilization, the electrodes were submerged in 70% ethanol for 10 min, washed with PBS without Ca^2+^Mg^2+^, and exposed for 30 min to UV light.

### 4.4. Cell Viability

To discard potential cytotoxic effects of EStim, cell viability was compared between the control and experimental groups by means of the alamarBlue reagent (Bio-Rad, Hercules, CA, USA), as performed in Leppik et al. (2022) [65]. Resazurin, the active ingredient of the alamarBlue reagent, is reduced to resorufin in the cytoplasms of viable cells. Briefly, after aspiration of culture medium, the cells were washed twice with sterile PBS without Ca^2+^Mg^2+^. A total of 1 mL of fresh medium was added to each well, along with 100 µL of alamarBlue reagent. A well containing only medium and alamarBlue reagent served as a blank control. The cells and blank control were then incubated for 4 h (37 °C, 5% CO_2_). The absorbance of the conditioned medium was measured at 570 and 600 nm using a plate reader (Tecan Infinite 200, Grodig, Austria). The percentage of alamarBlue reduction was calculated according to the following formula and molar extinction coefficient (E) values from the manufacturer’s protocol:$$\mathrm{Percentage}\;\mathrm{of}\;\mathrm{reduction}\;\mathrm{of}\;\mathrm{alamarBlue}=\frac{(O2\times A1)-(O1\times A2)\times 100}{\left(R1\times N2)-(R2\times N1\right)}$$
where the following holds true:O1 = E of oxidized alamarBlue at 570 nm (value: 80,586);O2 = E of oxidized alamarBlue at 600 nm (value: 117,216);R1 = E of reduced alamarBlue at 570 nm (value: 155,677);R2 = E of reduced alamarBlue at 600 nm (value: 14,652);A1 = absorbance of test wells at 570 nm;A2 = absorbance of test wells at 600 nm;N1 = absorbance of negative control well (medium plus alamarBlue but no cells) at 570 nm;N2 = absorbance of negative control well (medium plus alamarBlue but no cells) at 600 nm.

### 4.5. Recovery of Adherent Cells

After the determination of the cell viability, the cells were detached using Accutase solution (Sigma-Aldrich, St. Louis, MO, USA), collected using centrifugation, and resuspended in 100 µL of 0.5% BSA solution for manual counting with a hemocytometer (NanoEnTek, Gyeonggi-do, Republic of Korea) [64]. The total number of recovered cells was calculated for each well according to manufacturer’s protocol.

### 4.6. Cell Surface Marker Analysis Using Flow Cytometry

A total of 1 × 10^5^ cells in 100 µL of 0.5% BSA solution were incubated with a mixture of fluorochrome-conjugated antibodies (5 µL each: anti-human CD11b PE-Cy 5 (#555389), CD14 FITC (#561712), CD80 BV421 (#564160), CD86 APC (#555660), and CD206 PE (#555954)) or with a mixture of their corresponding fluorochrome-conjugated isotypes (5 µL each) as negative controls (all purchased from BD Biosciences, Heidelberg, Germany) for 30 min at 4 °C in the dark. The cells were then washed in 2 mL of 0.5% BSA solution using centrifugation (300× *g*, 5 min at 4 °C). The supernatant was discarded, and the pellet was vortexed and resuspended in 300 µL of 0.5% BSA solution. Flow cytometry was performed using a FACSCanto II flow cytometer (BD Biosciences, Franklin Lakes, NJ, USA) and analyzed with BD FACSDiva software version 6.1.3 (BD Biosciences, Franklin Lakes, NJ, USA). The events of interest were first gated by plotting the SSC area against the FSC area to exclude debris and very small particles. From this gate, the SSC height against the SSC area was plotted to identify and exclude cell doublets. Compensation was performed using unstained cells and single-stained controls to correct for spectral overlap [66]. Fluorescence data were acquired from at least 10,000 events in the viable cell gate.

### 4.7. Reverse Transcription Quantitative Polymerase Chain Reaction (RT-qPCR)

After removing the medium and non-adherent cells, the remaining cells were washed twice with sterile PBS without Ca^2+^Mg^2+^. RNA was then extracted by adding RLT lysis buffer (Qiagen, Hilden, Germany) directly to the adherent cells. The samples were subsequently processed using the RNeasy Mini Kit (Qiagen, Hilden, Germany), according to the manufacturer’s protocol. The RNA concentration and purity were measured using Nanodrop NanoVue Plus (Biochrom, Holliston, MA, USA). cDNA was synthesized from DNase-treated RNA using the iScript cDNA synthesis kit (Bio-Rad, Hercules, CA, USA). Gene expression in each sample was measured by means of RT-qPCR, with cDNA equivalent of 8 ng RNA and iTaq Universal SYBR Green Supermix (Bio-Rad, Hercules, CA, USA). All samples were amplified in a CFX96 Touch Real-Time PCR Detection System (Bio-Rad, Hercules, CA, USA) with commercial human gene-specific primers purchased from Qiagen (Hilden, Germany) or non-commercial primers designed with Primer-Blast tool of the National Center for Biotechnology Information (accessed date: 30 January 2022; https://www.ncbi.nlm.nih.gov/tools/primer-blast/) [67] and purchased from Sigma-Aldrich (Munich, Germany) or Eurofins Genomics Europe (Ebersberg, Germany). The sequences of the non-commercial primers are listed in Table 2. Melting curve analysis was performed to ensure the specificity of the amplification. GAPDH was used as a reference gene, and the gene expressions of target genes were calculated with the 2^−ΔΔCt^-method using the respective M0 control group as a reference [68].

### 4.8. Cytokines in Cell-Conditioned Medium

Cell supernatants stored at −80 °C were defrosted, and 4 samples of the same treatment from independent experiments were combined before measurement. Cytokine quantification was performed using a BD human inflammatory cytokine cytometric bead array (BD Biosciences, San Diego, CA, USA), simultaneously detecting human IL-8, IL-1β, IL-6, IL-10, TNF, and IL-12p70 proteins, according to the manufacturer’s protocol. The lower limits of cytokine detection were 3.6, 7.2, 2.5, 3.3, 3.7, and 1.9 pg/mL, respectively. Only for the detection of IL-8, the samples were diluted 1:20. The cytokine concentrations were measured using the FACSCalibur flow cytometer (BD Biosciences, San Jose, CA, USA) and calculated according to internal standard curves using FCAP Array v1.0 software (BD Biosciences, Franklin Lakes, NJ, USA) [61].

### 4.9. Statistical Analysis

Five independent experiments were conducted, each with three technical replicates (i.e., three wells) per group, except for the cytokine measurements, whose results come from three independent experiments. BiAS 11.12 software (Epsilon-Verlag, Darmstadt, Germany) was used for statistical assessment, while GraphPad Prism 8.0 software (GraphPad Software, Inc., San Diego, CA, USA) was employed for graphic design. All data sets were tested for normal distribution. Comparisons between the experimental and control groups were performed with the two-tailed Student’s t-test or Mann–Whitney non-parametric test, as appropriate, while the Kruskal–Wallis and Dunn’s multiple comparisons tests were used for the comparisons between the control groups. The results are presented as the mean values ± standard deviations or as box plots (box: borders are 25% and 75% quartiles; whiskers represent minimum and maximum values). In all cases, the significant difference between groups was considered when the *p* value was lower than 0.05.

## 5. Conclusions

In conclusion, our findings suggest that DC EStim modulates the polarization of THP-1-derived macrophages towards an M2 phenotype in both M0 and M1 cells. This is evidenced by its ability to counteract M1 features at the RNA and protein levels (surface markers and cytokines) and to induce the nuclear expression of M2 markers in M0 and M1 cells. Conversely, M2 cells appear to be less responsive to the EStim treatment employed in this study.

## Figures and Tables

**Figure 1 ijms-25-07272-f001:**
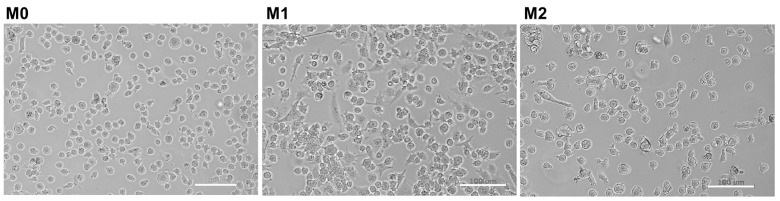
Morphology of THP-1-derived macrophages under different polarization states. Representative images of non-polarized macrophages (M0), pro-inflammatory macrophages (M1) induced with IFN-γ and LPS, and anti-inflammatory macrophages (M2) induced with IL-4 and IL-13 are shown. Cells were observed and imaged using phase contrast microscopy. Scale bar = 100 µm.

**Figure 2 ijms-25-07272-f002:**
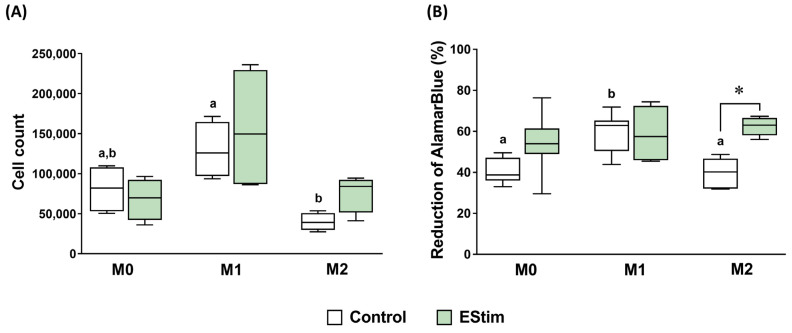
Effect of DC EStim on the number and viability of adherent THP-1-derived M0, M1 (IFN-γ + LPS), and M2 (IL-4 + IL-13) macrophages. Cell count (**A**) and viability, expressed as the percentage reduction of alamarBlue (**B**), were measured after priming with PMA (5 ng/mL) for 24 h, followed by a 24 h resting period and a 72 h polarization stage. During the polarization, cells were exposed (or not: control group) 1 h/day to 100 mV/mm of direct current electrical stimulation (EStim group). *n* = 5 independent experiments, each with 3 technical replicates. *: *p* < 0.05. Comparisons between control groups are shown with letters, where groups with the same letter are not significantly different.

**Figure 3 ijms-25-07272-f003:**
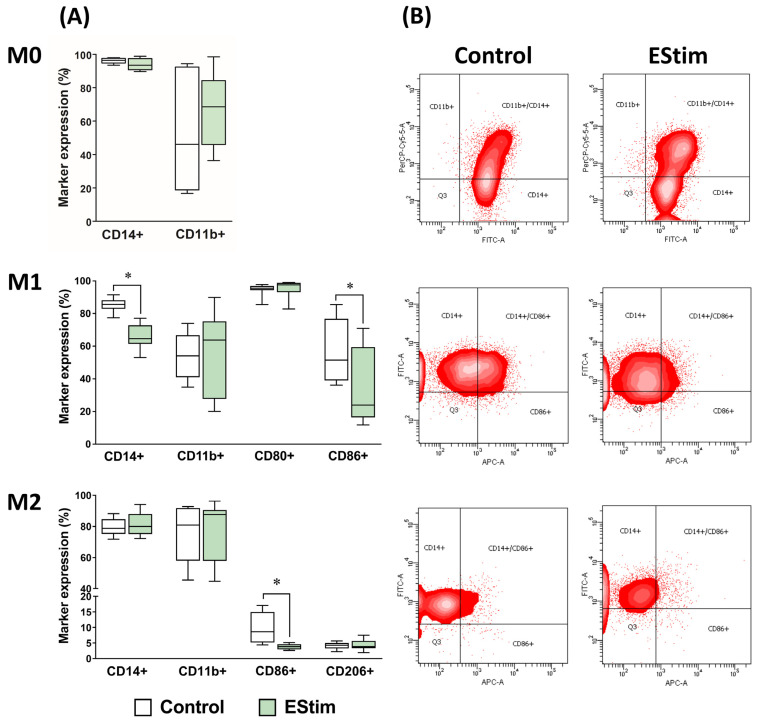
Effect of DC EStim on cell surface marker expression, assessed using flow cytometry in adherent THP-1-derived M0, M1 (IFN-γ + LPS), and M2 (IL-4 + IL-13) macrophages. Percentage of marker expression (**A**) and representative plots of the markers found to be significantly influenced by DC EStim (**B**). Cells were primed with PMA (5 ng/mL) for 24 h, followed by a 24 h resting period and a 72 h polarization stage. During polarization, cells were treated (or not: control group) 1 h per day with 100 mV/mm of direct current electrical stimulation (EStim group). A mix of 5 antibodies or their corresponding conjugates was employed for FACS analyses. Marker expressions lower than 1% were not included in the diagram. The corresponding isotype controls are shown in Appendix A. *n* = 5 independent experiments, each with 3 technical replicates. *: *p* < 0.05.

**Figure 4 ijms-25-07272-f004:**
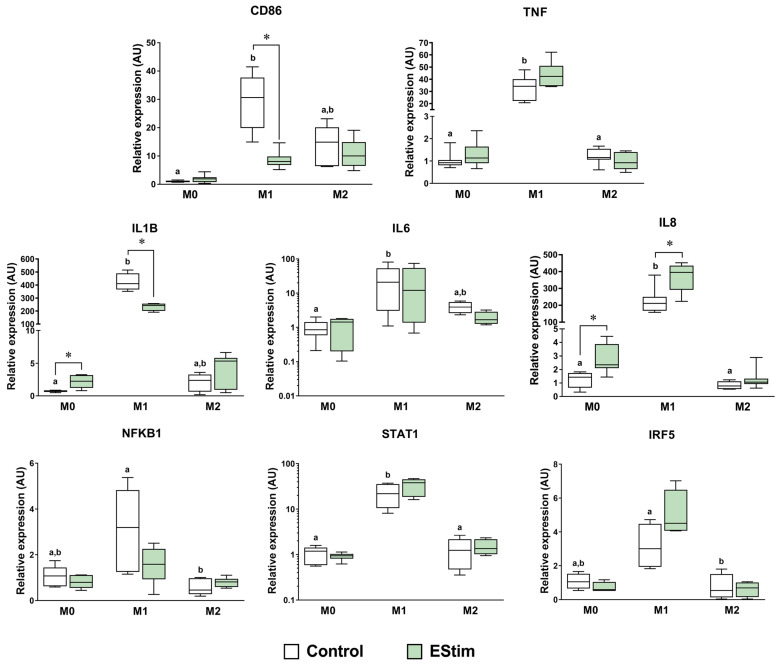
Gene expressions of M1-related markers and signaling mediators assessed using RT-qPCR in adherent THP-1-derived M0, M1 (IFN-γ + LPS), and M2 (IL-4 + IL-13) macrophages. Cells were primed with PMA (5 ng/mL) for 24 h, followed by a 24 h resting period and a 72 h polarization stage. During polarization, cells were treated (or not: control group) 1 h per day with 100 mV/mm of direct current electrical stimulation (EStim group). *n* = 5 independent experiments, each with 3 technical replicates. *: *p* < 0.05. Comparisons between control groups are shown with letters, where groups with the same letter are not significantly different.

**Figure 5 ijms-25-07272-f005:**
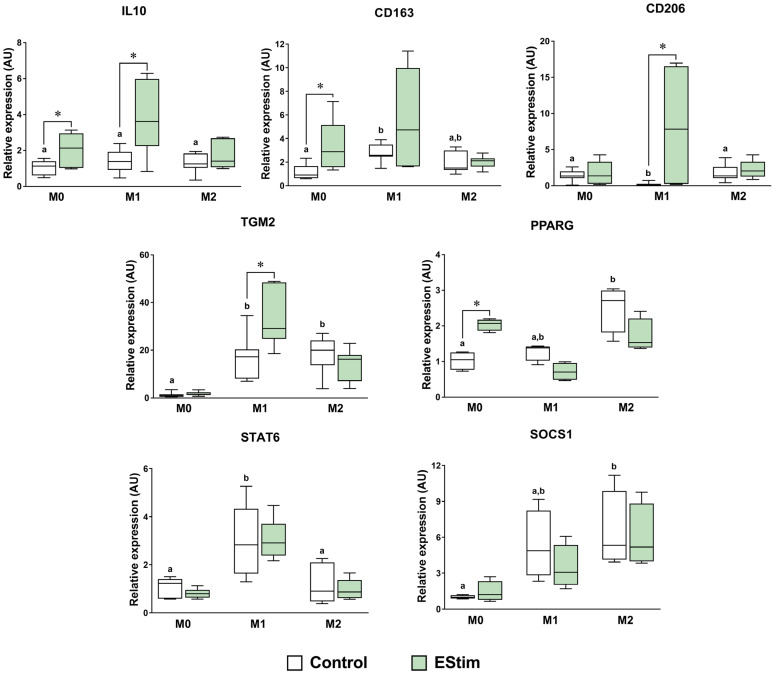
Gene expressions of M2-related markers and signaling mediators assessed using RT-qPCR in adherent THP-1-derived M0, M1 (IFN-γ + LPS), and M2 (IL-4 + IL-13) macrophages. Cells were primed with PMA (5 ng/mL) for 24 h, followed by a 24 h resting period and a 72 h polarization stage. During polarization, cells were treated (or not: control group) 1 h per day with 100 mV/mm of direct current electrical stimulation (EStim group). *n* = 5 independent experiments, each with 3 technical replicates. *: *p* < 0.05. Comparisons between control groups are shown with letters, where groups with the same letter are not significantly different.

**Figure 6 ijms-25-07272-f006:**
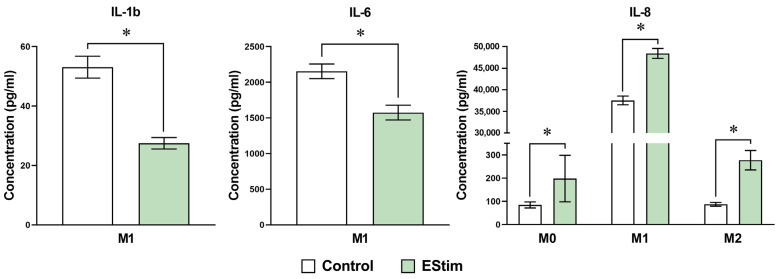
Cytokine levels assessed using cytokine cytometric bead array in cell supernatants of adherent THP-1-derived M0, M1 (IFN-γ + LPS), and M2 (IL-4 + IL-13) macrophages. Cells were primed with PMA (5 ng/mL) for 24 h, followed by a 24 h resting period and a 72 h polarization stage. During polarization, cells were treated (or not: control group) 1 h per day with 100 mV/mm of direct current electrical stimulation (EStim group). *n* = 3 independent experiments. Data are expressed as the mean ± SD. *: *p* < 0.05.

**Figure 7 ijms-25-07272-f007:**
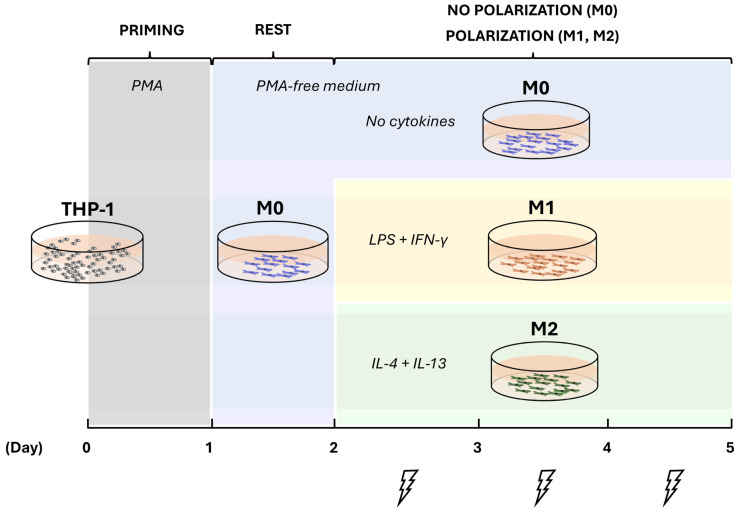
Experimental timeline. THP-1 cells were primed on day 0 with phorbol 12-myristate 13-acetate (PMA) for 24 h, left to rest for another 24 h, and then polarized (or not: M0) through cytokine treatments for three days. Direct current electrical stimulation was applied on experimental days 2, 3, and 4.

**Table 1 ijms-25-07272-t001:** Experimental setup.

Group	Priming (24 h)	Cytokines (Days 2–5)	DC EStim (Days 2–4)	Analysis (Day 5)
M0 control	PMA 5 ng/mL	-	-	Cell count, viability, gene and cell surface marker expressions, and cytokine secretion profiles.
M0 EStim	PMA 5 ng/mL	-	100 mV/mm, 1 h/day	
M1 control	PMA 5 ng/mL	IFN-γ 20 ng/mL, LPS 100 ng/mL	-	
M1 EStim	PMA 5 ng/mL	IFN-γ 20 ng/mL, LPS 100 ng/mL	100 mV/mm, 1 h/day	
M2 control	PMA 5 ng/mL	IL-4 20 ng/mL, IL-13 20 ng/mL	-	
M2 EStim	PMA 5 ng/mL	IL-4 20 ng/mL, IL-13 20 ng/mL	100 mV/mm, 1 h/day	

PMA: phorbol 12-myristate 13-acetate; IFN-γ: interferon gamma; LPS: lipopolysaccharide; IL: interleukin; DC EStim: direct current electrical stimulation.

**Table 2 ijms-25-07272-t002:** Sequences of non-commercial RT-qPCR primers.

Gene	Forward (5′->3′)	Reverse (5′->3′)	Annealing Temperature (°C)	Amplicon Size
*CD163*	GCCATTCTGAGCCACACTGA	AGTCCAGGTCTTCATCAAGGT	61	87
*CD206*	CGATCCGACCCTTCCTTGAC	TGTCTCCGCTTCATGCCATT	63	120
*IL1B*	CAGAAGTACCTGAGCTCGCC	AGATTCGTAGCTGGATGCCG	62	153
*IL8*	AAGGAACCATCTCACTGTGTGTAAAC	ATCAGGAAGGCTGCCAAGAG	62	70
*IRF5*	CCAGCCAGGACGGAGATAAC	AGGTTGGCCTTCCACTTGG	56	106
*NFKB1*	GTTTGTCCAGCTTCGGAGGA	CACTACCACCGCCGAAACTA	59	149
*SOCS1*	TTTTCGCCCTTAGCGTGAA	CATCCAGGTGAAAGCGGC	56	81
*STAT1*	GGAAATCAGACAGTACCTGGCA	ACAGGAGGTCATGAAAACGGA	56	101
*STAT6*	AAAGTGCAGCGGCTCTATGT	GGTGCTGGACAGTGTCTGAA	57	151
*TNF*	CCAGGCAGTCAGATCATCTTCTC	AGCTGGTTATCTCTCAGCTCCAC	62	150

## Data Availability

The data presented in this study are available on request from the corresponding author.

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
