# Peer review of "Direct Current Electrical Stimulation Shifts THP-1-Derived Macrophage Polarization towards Pro-Regenerative M2 Phenotype"

_ijms, 2024, doi:10.3390/ijms25137272_

Round 1

Reviewer 1 Report

Comments and Suggestions for Authors

Summary:

In this original article, Bianconi et al., studied macrophage (Mo) polarization, which was assessed through gene and surface marker expression, and cytokine secretion profile. Following direct current, eletrical stimulation (DC  EStim) treatment, non-stimulated cells exhibited upregulation of M2 marker genes, such as IL10, CD163, and PPARG. In M1 cells, DCEStim upregulated gene expression of M2 markers IL10, TGM2, and CD206, and down- regulated M1 marker gene CD86. EStim treatment also reduced surface expression of CD86 and secretion of pro-inflammatory cytokines IL-1β and IL-6. The results presented by the authors suggest that DC EStim exerts differentially pro-M2 effects depending on macrophage phenotype: it upregulates typical M2 genes in M0 and M1 cells while inhibiting M1 marker CD86 at nuclear and protein level and the secretion of proinflammatory interleukins in M1 cells. Conversely, M2 cells appear to be less responsive to the EStim treatment employed in this study.

The article is interesting, and covers an important topic in immunobiology. However, I have some questions and suggestions before the article is accepted for publication

MAJOR:

In this article, the authors used macrophages derived from the THP-1 monocytic line, which is derived from a patient with acute monocytic leukemia. Although it is used as a model to study the biology of macrophages, it is already known that there are marked differences in relation to human macrophages derived from monocytes isolated from peripheral blood. Would the authors like to discuss this issue? In addition, if possible, it would be interesting to carry out some assays with macrophages derived from blood monocytes.

I suggest the authors rewrite the captions for all figures . The figures present a lot of information, but the captions are poor in content. They need to be rewritten in more detail.

This article cover an interesting topic about electrical stimulation. Over the last few years, several papers have demonstrated and discussed how electrical stimulation may be employed for regulating the behavior of different immune cells. Examples include macrophages, T and B cells, and neutrophils.  A better understanding of how these strategies influence the polarization, phagocytosis, migration, and differentiation of immune cells is of great significance in the field. In my opinion, the authors should discuss this topic in more depth to enrich the manuscript and facilitate understanding for readers.

Regarding materials and methods, I suggest the authors describe in more detail the methodology carried out for each experiment. In addition, if the methods used have been previously published by the same group or by other research groups, please, at the end of each methodology, add the reference of the original paper.

MINOR:

Minor editing of English language required.

Comments on the Quality of English Language

Minor editing of English language required.

Reviewer 2 Report

Comments and Suggestions for Authors

Review on Bianconi s. et.al. “Direct Current Electrical Stimulation Shifts THP-1-Derived Macrophage Polarization Towards Pro-Regenerative M2 Phenotype”

This article presents interesting findings on how DC Estim can affect M1 and M2 polarization. However, it lacks numerous validation experiments (See comments below). Additionally, the authors should evaluate key M1 and M2-related transcription factors or signaling pathways, such as STAT, to provide some synopsis of molecular mechanisms behind DC Estim and M2 polarization.

Figure 1: Ensure all images, including M0 macrophages, are taken with the same phase contrast settings.

Figure Citations: Reorder the figures or modify the text to ensure citations are in sequential order.

Figure Legends: Indicate the type of results (Flow cytometry, RT-qPCR, Western blot, ELISA) for each panel.

Representative FACS Plots: Include representative FACS plots in the main figures along with quantified results.

Figures 4, 5, and 6: Validate cytokine changes using at least two methods, such as cell cytometry and RNA expression, ELISA and RNA expression, Western blot and RNA expression, or a combination of these techniques.

Table 1: Clarify whether IL-4 or IL-14 was used.

Figure 8: Move to the supplemental section or place it before the first DC Estim data presentation. Ensure all figures are explained in chronological order and cited at least once in the main text results section.

Round 2

Reviewer 1 Report

Comments and Suggestions for Authors

I thank the authors for their efforts to improve the manuscript 

Reviewer 2 Report

Comments and Suggestions for Authors

The authors have answered all the review questions appropriately and improved the manuscript accordingly.

Comments on the Quality of English Language

English is okay